# Research on the Anti-Fatigue Effects and Mechanisms of Arecoline in Sleep-Deprived Mice

**DOI:** 10.3390/nu16162783

**Published:** 2024-08-21

**Authors:** Danyang Wang, Yuan Sun, Jiameng Liu, Jing Sun, Bei Fan, Cong Lu, Fengzhong Wang

**Affiliations:** 1Institute of Food Science and Technology, Chinese Academy of Agricultural Sciences (CAAS), Beijing 100193, China; wdanyang1111@163.com (D.W.); sy061998@163.com (Y.S.); ljiam88@163.com (J.L.); ycsunjing2008@126.com (J.S.); fanbei517@163.com (B.F.); 2College of Food Science and Engineering, Shanxi Agricultural University, Jinzhong 030801, China; 3National Nanfan Research Institute (Sanya), Chinese Academy of Agricultural Sciences, Sanya 572024, China

**Keywords:** arecoline, anti-fatigue, glycolipid metabolism, oxidative stress, neurotransmitter

## Abstract

The betel nut is one of the most widely consumed addictive substances in the world after nicotine, ethanol, and caffeine. Arecoline is an active ingredient from the areca nut. It has many pharmacological effects and can affect the central nervous system. In this study, we found that arecoline can relieve fatigue behavior. Objective: This research aims to estimate the anti-fatigue effects of arecoline and explore its underlying mechanisms using a murine model of central fatigue precipitated by sleep deprivation (SD). Methods: Seventy-two male C57BL/6 mice were randomly assigned to six groups: a control group, an SD-induced fatigue model group, a group that received Rhodiola Rosea capsules (2.5 mg/kg), and three arecoline groups, which were administered at low, medium, and high doses (10, 20, and 40 mg/kg, respectively). Following 28 days of continuous administrations, the effects of arecoline on mouse fatigue-related behaviors were assessed by behavioral tests, including grip strength, rotarod performance, and weight-bearing swimming endurance. The release levels of the related biochemical markers were measured by enzyme-linked immunosorbent assays (ELISAs). Western blotting was employed to quantify the expression levels of nuclear factor erythroid 2-related factor (Nrf2), Kelch-like ECH-associated protein 1 (Keap1), heme oxygenase 1 (HO-1), sequestosome-1 (p62), and NADPH quinone oxidoreductase 1 (NQO1) in the gastrocnemius muscle. Results: Arecoline administration notably enhanced grip strength, delayed the onset of fatigue as evidenced by extended latencies in rotarod tests, and increased the duration of weight-bearing swimming in mice. In the elevated plus maze, arecoline obviously decreased both the number of entries and the total distance traveled in the open arms. Arecoline markedly decreased the contents of creatine kinase, blood urea nitrogen, lactate dehydrogenase, triglycerides, and cholesterol in the serum, while it elevated the levels of total testosterone, lactate dehydrogenase, and immunoglobulin G. Furthermore, it significantly increased the activities of superoxide dismutase (SOD), catalase (CAT), and glutathione peroxidase in the gastrocnemius muscle, reduced malondialdehyde levels, augmented hippocampal SOD and CAT activity, and elevated glycogen stores in both liver and muscle tissues. Neurotransmitter levels showed significant increases, cytokine levels were markedly reduced, and the expressions of Nrf2, Keap1, NQO1, p62, and HO-1 in brain tissues were significantly upregulated. Conclusions: This study demonstrates that arecoline has anti-fatigue activity, and the specific mechanisms are associated with elevating glucose and lipid metabolism levels, relieving oxidative stress damage, inhibiting neuroinflammatory response, and regulating neurotransmitter levels and the Keap1/Nrf2/HO-1 signaling pathway. The research provides a new direction for arecoline’s potential in preventing and improving fatigue.

## 1. Introduction

Fatigue is defined as the inability of an organic physiological process to maintain its function at a specific level and/or to maintain a predetermined intensity of exercise [1]. Fatigue emerges as a predominant symptom of sub-health, representing a pre-disease condition and a harbinger of potential pathologies. It triggers a cascade of physiological responses including sleep deprivation, endocrine imbalances, metabolic dysfunctions, and immune impairments. The modern lifestyle, characterized by fast-paced living, stress, unhealthy dietary habits, irregular routines, sleep deprivation, sedentary behavior, and persistent negative emotions, exacerbates the prevalence of sub-health conditions. Chronic fatigue can eventually lead to serious health problems such as accelerated aging, anxiety, depression, and neurodegenerative diseases, such as cancer and Parkinson’s, significantly reducing an individual’s quality of life and productivity at work [2]. Research shows that 12% of middle school students in China suffer from chronic fatigue [3]. The current pharmacological solutions, such as cerebral cortex stimulants, can offer temporary relief of fatigue, while their potential for adverse effects like convulsions, mental disturbances, and dependency restricts their utility. Therefore, it is urgent to seek more safe and effective anti-fatigue compounds and elucidate their mechanisms of action for developing anti-fatigue nutritional health products to meet the nutritional needs of modern people.

Food and drug homologous substances have the characteristics of both medicine and food, which is a special resource in China and has great potential for exploitation. With the increase in health awareness and the pursuit of health, food with dual medicinal and nutritional value caters to the preferences of modern people. *Areca catechu* L. is an evergreen tree of the Areca genus of the palm family (Palmaceae) and one of the cash crops in the south tropical and subtropical areas of China [4]. The dried, mature seeds are called areca nuts. Fresh fruits and dried processed products are its main food routes. In addition, *Areca catechu* L. has a wide array of bioactive actions, such as anti-inflammatory [5], antiviral [6], antioxidant [4], anti-aging [7], antidepressant [8], anti-hypoxic [9], and antithrombotic properties, and aids in blood sugar reduction [10,11], vascular protection, and gastrointestinal enhancement [12], making it highly esteemed as one of the “four southern medicines” of China. As important active substances in areca nuts, alkaloids exhibit potent biological actions, particularly on the nervous and digestive systems [13,14,15]. The primary alkaloids, including arecoline and its derivatives, constitute approximately 0.3% to 0.7% of the total content [16,17]. It has been reported that the extract of Areca catechu L can enhance cognitive function, elevate the antioxidant capacity of brain tissue, and reduce the signs of tissue aging in mice, showing anti-aging effects [18]. Furthermore, our previous findings showed that arecoline has neuroprotective activities both in the lipopolysaccharide (LPS)-induced BV2 inflammatory response in mouse microglia [19] and H_2_O_2_-induced oxidative stress-damaged model in SH-SY5Y cells, elucidating its mechanisms related to the antioxidation and anti-apoptosis [20]. However, the anti-fatigue effect of arecoline has not yet been reported. In order to further explore the biological activity of arecoline and elucidate its mechanisms, the present study was conducted to determine whether arecoline could improve the fatigue induced by sleep deprivation by modulating the antioxidant system and attenuating the inflammatory response.

## 2. Materials and Methods

### 2.1. Materials

Arecoline (63-75-2, no. A14660) was obtained from Shanghai Jizhi Biochemical Technology Co., Ltd. (Shanghai, China). Rhodiola Rosea capsules were provided by Shaanxi Jiahe Phytochemistry Co., Ltd. (Xi’an, Shaanxi, China). Commercial sandwich enzyme-linked immunosorbent assay (ELISA) kits for gamma-aminobutyric acid (GABA), dopamine (DA), norepinephrine (NE), acetylcholine (Ach), 5-hydroxytryptamine (5-HT), tumor necrosis factor (TNF-α), interleukin-1β (IL-1β), and interleukin-6 (IL-6) were obtained from Jiancheng Biological Technology Co., Ltd. (Nanjing, Jiangsu, China). The commercial kits for serum testosterone (TTS), corticosterone (CTC), lactic acid (LD), triglyceride (TG), blood urea nitrogen (BUN), creatine kinase (CK), immunoglobulin G (IgG), liver glycogen (LG), muscle glycogen (MG), superoxide dismutase (SOD), glutathione peroxidase (GSH-Px), malondialdehyde (MDA), and catalase (CAT) were purchased from Jianglai Biological Technology Co., Ltd. (Shanghai, China). Antibodies targeting nuclear factor erythroid 2-related factor (Nrf2), Kelch-like ECH-associated protein 1 (Keap1), heme oxygenase 1 (HO-1), sequestosome-1 (p62), and NADPH quinone oxidoreductase 1 (NQO1) were supplied by Cell Signaling Technology (Boston, MA, USA). HRP-labeled goat anti-rabbit (111-035-003) and HRP-labeled goat anti-mouse (115-035-003) were purchased from The Jackson Laboratory.

### 2.2. Ethical Statement

The study was performed in compliance with the National Institutes of Health and agency guidelines for the Care and Use of Laboratory Animals and under the approval and supervision of the Animal Ethics Committee at the Institute of Food Science and Technology, Chinese Academy of Agricultural Sciences (approval number: SYXK-2023100312).

### 2.3. Animals and Experimental Design

Seventy-two male C57BL/6 mice, weighing between 18.0 and 22.0 g, were acquired from Beijing Weitong Lihua Laboratory Animal Technology Co., Ltd. (Beijing, China), holding an experimental animal production license (SCXK (Guangdong) 2019-0063) and an experimental animal qualification certificate (no. 44829700011576). Before the tests, mice were given a week to adjust to their new environment within the animal facility, which maintained conditions of a 12-h light and dark cycle, a stable temperature ranging from 23 to 25 °C, and a relative humidity of approximately 55–65%. They had unrestricted access to food and water throughout this period. Based on body weight, the mice were divided into 6 groups (n = 12 in each group): (1) the control (not subjected to any stress, distilled water); (2) the model (SD procedure, distilled water); (3) Rhodiola (SD procedure, 2.5 mg/kg Rhodiola Rosea capsules); (4) L-Arecoline (SD, 10 mg/kg Arecoline); (5) M-Arecoline (SD, 20 mg/kg Arecoline); and (6) H-Arecoline (SD, 40 mg/kg Arecoline). The mice were given the corresponding solution 1 h before the start of the daily SD procedure and behavioral tests. All drugs were dissolved in distilled water and administered intragastrically at 20 mL/kg body weight. Following 28 days of SD procedure and continuous treatment, the forelimb grip strength, rotary latency, and exhaustive swimming tests were conducted to assess the anti-fatigue effects of arecoline. After the behavioral tests, the mice were killed.

### 2.4. SD Procedure

The SD procedure was performed as previously described [21]. In brief, the mice, except for the control group, were acclimated to the automated sleep interruption apparatus (SIA) for 3 h (from 8:00 a.m. to 11:00 a.m. daily), which lasted for 3 days before the SD procedure; meanwhile, the mice in the control group were placed in a static SIA copy apparatus. Then, a 28-day period of SD commenced, wherein SD mice remained in the SIA continuously for 24 h each day, while control mice continued to inhabit the static SIA copy apparatus. At the same time, the mice were administrated daily with different doses of arecoline or Rhodiola by oral gavage, whereas the control and SD model groups received the corresponding volume of solvent.

### 2.5. Behavioral Tests

#### 2.5.1. Forelimb Grip Strength Test

On the 29th day, the forelimb grip strength of the mice was measured by a low-force analysis system. In brief, the procedure involved having the mice grip the available grids in the instrument with their front paws. Subsequently, their tails were elevated and gently pulled backward, enabling the maximum force exerted by the mice to be automatically recorded by the system. The forelimb grip strength for each mouse was determined by calculating the average of three consecutive measurements, ensuring the accuracy and reliability of the data collected. This method provided a quantitative assessment of the muscular strength and fatigue resistance in the mice following the experimental interventions.

#### 2.5.2. Rotary Latency Experiment

The rotarod experiment was conducted to evaluate exercise endurance and motor coordination in mice. In the test, mice were positioned on a cylindrical rod that rotated at a gradually increasing speed. The procedure involved closely observing the performance of the mice as the rotation speed escalated, meticulously noting endurance time and the specific speeds at which the mice could no longer maintain their balance and subsequently fall off the rod. At the commencement of the test, the rod began to rotate, with the experimenter tasked with monitoring the behavior of the mice on this apparatus. The precise moment a mouse fell or failed to maintain balance, indicating a loss of motor coordination or endurance, was recorded as the drop time. To ensure a comprehensive and reliable assessment, the test was repeated multiple times, with each trial lasting for several minutes. This repetitive testing approach allowed the determination of the key parameters of the mice, such as endurance and overall exercise capacity, providing valuable insights into their physical performance and motor skills post-experimental interventions.

#### 2.5.3. Exhaustive Swimming Test

The swimming endurance of the mice was evaluated using an adjustable current swimming pool, with some modifications to the standard protocol. Briefly, 30 min following the final administration via gavage, each mouse was equipped with a lead sheath, equivalent to 5% of its body weight, affixed to the root of its tail. Subsequently, each mouse was placed individually into an acrylic plastic pool, which was maintained at a water temperature of 25 °C and a depth of 35 cm. The exhaustive swimming time, serving as the primary measure of endurance, was meticulously recorded from the moment the mouse began swimming until it could no longer resurface within a 10-s interval. This assessment method effectively quantifies the physical stamina and fatigue resistance of the mice, offering insights into their endurance capabilities under stress-induced conditions.

### 2.6. Sample Collection

Immediately following the conclusion of the final exhaustive swim test, each mouse was euthanized, and blood samples were extracted from the eyeball. The serum was obtained by centrifugation of these samples at 4 °C and 3500 rpm for 10 min. Key organs, including the liver, hippocampus, and gastrocnemius muscle, were excised. The liver and muscle tissues were kept at −80 °C for further analysis.

### 2.7. Biochemical Parameter Assays

The activities of serum testosterone (TTS), corticosterone (CTC), lactic acid (LD), triglyceride (TG), blood urea nitrogen (BUN), creatine kinase (CK), and immunoglobulin G (IgG) in serum were detected. The activities of liver glycogen (LG) in the liver and muscle glycogen (MG) in gastrocnemius were determined. And the levels of superoxide dismutase (SOD), glutathione peroxidase (GSH-Px), malondialdehyde (MDA), and catalase (CAT) in the muscle and hippocampus were determined using commercial assay kits (Jianglai Biological Technology Co., Ltd., Shanghai, China), according to the manufacturer’s instructions.

The hippocampus sample was collected to measure the levels of 5-hydroxytryptamine (5-HT), dopamine (DA), norepinephrine (NE), acetylcholine (Ach), gamma-aminobutyric acid (GABA), tumor necrosis factor (TNF-α), interleukin-1β (IL-1β), and interleukin-6 (IL-6) using commercial ELISA kits (Jiancheng Biological Technology Co., Ltd., Nanjing, Jiangsu, China) based on the manufacturer’s instructions.

### 2.8. Western Blotting Analysis

Protein extraction from the gastrocnemius muscle was performed using lysis buffer, supplemented with a phosphatase inhibitor, to ensure comprehensive protein recovery while preventing dephosphorylation. Following lysis, the mixture was centrifuged at 4 °C for 15 min at 12,000 rpm, enabling the collection of the supernatant, which contains the protein content. For protein analysis, equal quantities of the protein samples (30 μg) were subjected to electrophoresis on an 8% SDS-PAGE to achieve separation based on molecular weight. Subsequently, the separated proteins were transferred onto PVDF membranes, a step crucial for subsequent immunoblotting. To prevent non-specific binding, the membranes were incubated with 5% (*w*/*v*) skimmed milk at room temperature, serving as a blocking agent. The primary antibody was then incubated overnight at 4 °C, ensuring optimal binding affinity and specificity. Following this incubation, the membranes were washed three times with T-TBS to remove unbound antibodies, and then with goat anti-rabbit IgG (H + L) HRP 1:10,000, incubated at room temperature for 1 h. These secondary antibodies were conjugated to an enzyme that produced a detectable signal upon substrate addition, allowing for visualization of the protein bands. Blot development was conducted using a Gel Image Analysis System, which facilitated the visualization and quantification of the protein bands. For quantitative analysis, the relative protein expression levels were normalized to GAPDH, a housekeeping protein, using ImageJ software (version 6.0). This normalization is critical for accounting for variations in sample loading and transfer efficiency, providing a reliable measure of the protein levels of interest in the context of the study’s experimental conditions.

### 2.9. Statistical Analysis

Statistical analyses were conducted via SPSS 25.0 software and the results were expressed as mean ± SEM. One-way analysis of variance followed by Fisher’s LSD post hoc tests was carried out to compare the groups. The results of all tests are considered to be statistically significant when *p* < 0.05.

## 3. Results

### 3.1. Effect of Arecoline on Fatigue-Related Behaviors in SD-Induced Mice

The experimental results, summarized in Figure 1, elucidate the effects of arecoline on the physical performance of mice subjected to a regimen of grip strength, rotarod endurance, and weight-bearing swimming tests. Prior to the treatments, the grip strength of SD model mice, as well as those treated with Rhodiola and Aprodine, was markedly lower when compared to that of the control group (*p* < 0.01, Figure 1A), indicating an induced state of fatigue. Meanwhile, there were no obvious differences among the various treatment groups at this initial stage. Post-treatment observations revealed a marked decrease in grip strength within the model group compared to the control group (*p* < 0.01, Figure 1B), highlighting the persistence of fatigue. Conversely, treatment with arecoline at varying dosages resulted in a significant increase in grip strength across all arecoline-dosed groups when compared to the model group (*p* < 0.05, *p* < 0.01, Figure 1B), demonstrating arecoline’s potential in mitigating fatigue and enhancing muscular strength. In the rod-turning test, latency times for the model group were significantly less than those of the control group (*p* < 0.01, Figure 1C), suggesting impaired motor coordination or endurance due to induced fatigue. However, treatments with arecoline showed a trend toward increased latency times in all arecoline-dosed groups compared to the model group, indicating an improvement in motor coordination and endurance. The weight-bearing swimming test results further supported arecoline’s anti-fatigue effects. Compared to the control group, the model group exhibited a significant reduction in swimming time (*p* < 0.05, Figure 1D), indicative of decreased endurance. Notably, the LA group demonstrated a significant increase in weight-bearing swimming time when compared with the model group (*p* < 0.05, Figure 1D), affirming arecoline’s efficacy in enhancing endurance and resistance to fatigue.

### 3.2. Effects of Arecoline on Serum Biochemical Indexes in Mice

LD and BUN are products of glycolysis and amino acid metabolism, while CK is a key indicator that reflects fatigue and assesses muscle cell damage. As shown in Figure 2, when compared with those of the control group, the levels of CTC, BUN, LD, TG, and CK in the serum of SD model mice were significantly increased (*p* < 0.05, *p* < 0.01, *p* < 0.001, Figure 2A,C–F), at the same time, the activities of TTS and IgG were significantly decreased (*p* < 0.001, Figure 2B,G). However, administrations with Rhodiola and arecoline groups significantly decreased the serum levels of CTC, BUN, LD, TG, and CK (*p* < 0.001, *p* < 0.0001, Figure 2A,C–F), and elevated the TTS level and IgG activity (*p* < 0.0001, Figure 2B,G). These results suggest that arecoline intervention can significantly increase the serum antioxidant level of model mice.

### 3.3. Effects of Arecoline on the Oxidative Stress Indexes of Gastrocnemius Muscle in Mice

Free radicals and oxidative damage play a key role in excessive fatigue. The fatigue indexes of gastrocnemius in mice are shown in Figure 3. Compared with the control, MDA in the model group was significantly increased (*p* < 0.0001, Figure 2A), and SOD, CAT, and GSH-Px activities were markedly declined (*p* < 0.01, *p* < 0.0001, Figure 3B–D). Compared with the model group, MDA levels in arecoline dose groups were significantly decreased (*p* < 0.01, *p* < 0.0001, Figure 3A), and SOD, CAT, and GSH-Px levels in arecoline dose groups showed significant increases (*p* < 0.05, *p* < 0.01, *p* < 0.001, *p* < 0.0001, Figure 3B–D). The results showed that arecoline intervention significantly improved the antioxidant levels in the gastrocnemius of model mice.

### 3.4. Effects of Arecoline on the Glycolipid Metabolism Indexes in Mice

To elevate the effects of arecoline administration on glucose and lipid metabolism, the levels of liver glycogen and muscle glycogen in mice were measured. Both the liver glycogen level and muscle glycogen level were significantly decreased in model mice when compared to those of the control group (*p* < 0.001, Figure 4A,B). Meanwhile, administrations with arecoline and Rhodiola all significantly increased the contents of liver glycogen and muscle glycogen of SD-induced mice (*p* < 0.01 or *p* < 0.0001, Figure 4A,B). These results showed that arecoline intervened with liver glycogen content and muscle glycogen content in model mice, achieving anti-fatigue effects.

### 3.5. Effects of Arecoline on Biochemical Indices of Mouse Hippocampus

#### 3.5.1. Effects of Arecoline on the Oxidative Stress Indexes in the Hippocampus of Mice

Compared with the control group, SOD and CAT activities in the model group were significantly reduced (*p* < 0.001, Figure 5A,B). Compared with the model group, the SOD activities of mice in low-, medium-, and high-dose arecoline groups were markedly increased (*p* < 0.05, *p* < 0.01, or *p* < 0.001, Figure 5A), and the CAT activities of mice in medium- and high-dose arecoline groups were significantly increased (*p* < 0.05, Figure 5B).

#### 3.5.2. Effects of Arecoline on Cytokines TNF-α, IL-6 and IL-1β in Mouse Hippocampus

SD procedure significantly elevated the levels of TNF-α, IL-6, and IL-1β in the hippocampus of model mice when compared to those of the control group (*p* < 0.001, *p* < 0.0001, Figure 6A–C). However, IL-1β content in medium and high doses of arecoline mice was both significantly decreased (*p* < 0.05, Figure 6C), and TNF-α and IL-6 contents in low, medium, and high doses of arecoline mice were all significantly decreased (*p* < 0.001, *p* < 0.0001, Figure 6A,B). The results show that arecoline administrations can significantly inhibit the oxidative stress level induced by SD.

#### 3.5.3. Effects of Arecoline on the Content of 5-HT, DA, NE, GABA, and Ach in the Hippocampus of Mice

Compared with the control group, 5-HT, DA, NE, GABA, and Ach in the model group were significantly decreased (*p* < 0.05, *p* < 0.01, *p* < 0.001, *p* < 0.0001, Figure 7A–E). The contents of DA, NE, 5-HT, and GABA in low, medium, and high doses of arecoline were significantly increased (*p* < 0.01, *p* < 0.001, *p* < 0.0001, Figure 7A–D). The results showed that arecoline intervention markedly promoted neurotransmitter release in the SD mouse hippocampus. The results showed that arecoline intervention reduced the level of cytokines in the hippocampus of the model mice to achieve anti-fatigue effects.

### 3.6. Effects of Arecoline on the Keap1/Nrf2/HO-1 Signaling Pathway in the Gastrocnemius of Mice

The Keap1/Nrf2/HO-1 signaling pathway is believed to be critical under oxidative stress conditions. As shown in Figure 8A–C, compared with the control, the expressions of Nrf2, P62, Keap1, HO-1, and NQO1 proteins in the model group were obviously downregulated. However, administrations with Rhodiola and arecoline significantly up-regulated the expressions of Nrf2, P62, Keap1, HO-1, and NQO1 (*p* < 0.01, *p* < 0.001, *p* < 0.0001, Figure 8B–F).

## 4. Discussion

Sleep is a vital physiological process for humans and a protective mechanism for maintaining autonomic nervous system and immune system homeostasis [22]. Shorter sleep times can lead to fatigue, increased mood swings, and affect cognition. It is well-known that a lack of sleep can lead to several significant effects on the cardiovascular, endocrine, immune, and nervous systems [23,24]. Fatigue is a long-term or temporary state of physical or mental weakness and exhaustion, often accompanied by a lack of energy, motivation, and focus. The SD-induced fatigue model can mimic a state of fatigue induced by sleep deprivation and is used to screen the candidate anti-fatigue active substances. Grip test, rod spinning test, and weight exhaustion swimming tests were conducted in the current study, which are the classic behavioral paradigms used for evaluating fatigue-like behaviors and the body strength of experimental animals [25,26]. The results of the current study showed that after 14 days of chronic sleep deprivation, the grasping power, rod turning time, and weight-bearing swimming time of mice were significantly reduced, showing that the fatigue model in mice was successfully established. Meanwhile, administrations with arecoline (10, 20, and 40 mg/kg) remarkably improved these behavioral changes in SD-induced mouse fatigue, indicating remarkable anti-fatigue action.

Oxidative stress represents a state in which the level of reactive oxygen species (ROS) in the body is dramatically elevated. Under normal circumstances, the ROS production and antioxidant systems in the body are in a state of dynamic balance. Meanwhile, excessive ROS produced by strenuous exercise can result in an imbalance in the oxidative stress response, resulting in the lipid peroxidation of membrane structures and the formation of the lipid peroxidation product MDA, which changes the fluidity and permeability of cell membranes, damages skeletal muscle and liver mitochondria, and ultimately leads to fatigue. GSH-Px is an endogenous antioxidant enzyme that plays a key role in maintaining the balance of antioxidant stress in the body [27]. Studies have found that after continuous administration of areca nuts for 14 days, the activities of SOD and CAT in mice increased significantly and the expression of MDA decreased significantly, indicating that areca nuts had good antioxidant activity [28]. The results of this study showed that arecoline groups could significantly reduce the MDA content of mice, and significantly increase the expression levels of SOD, CAT, and GSH-Px in mice, which was consistent with the results. The results showed that arecoline could relieve fatigue by alleviating oxidative stress damage.

Previous studies have shown that CTC, TTC, BUN, LD, TG, and CK levels are closely related to exercise [29]. High-intensity exercise accelerates the accumulation of LD. This study showed that arecoline can reduce the contents of serum CTC, BUN, LD, TG, and CK, and finally play an anti-fatigue role, providing sufficient evidence to prove that arecoline can regulate fatigue-related biochemical indicators and reduce muscle damage. The body’s main source of energy for movement is glycogen, which is stored in the liver and muscles. Glycogen provides energy through peroxidation during high-intensity exercise [30,31]. Therefore, glycogen reserve can directly improve exercise ability and slow down the occurrence of exercise fatigue [32]. The contents of liver glycogen and muscle glycogen in the arecoline group were significantly higher than those in the model group. The results showed that arecoline was beneficial in delaying energy consumption and had a significant anti-fatigue effect.

Arecoline regulates levels of neurotransmitters in the brain [33]. It has been reported that the gut microbiota affects the brain through a variety of pathways, including changes in the gamma-aminobutyrate (GABA) system, and contributes to a variety of mental disorders [34]. GABA is a major inhibitory neurotransmitter in the central nervous system. GABA is a major molecule that regulates sleep and immune responses. It is found in extremely high levels in the brain, especially in the hypothalamus, where about 30% of synapses are GABA transmitters [35]. It has been reported that areca effectively inhibits central GABA uptake in rat brain sections [36]. In addition, the alkaloids in betel nuts are potent depressants for GABA uptake, and chewing betel nuts may affect sympathetic nerve function and increase the plasma concentration of norepinephrine [37]. Ach was the first neurotransmitter to be discovered and has multiple key roles in both the peripheral and central nervous systems [38]. Arecoline has discriminative stimulative effects associated with muscarinic and nicotinic receptors [39]. The results of this study showed that arecoline interventions reversed the decline in 5-HT, DA, NE, GABA, and Ach levels, suggesting that the improvement of the cholinergic system function is one of the potential mechanisms of arecoline’s anti-fatigue activity. SD can activate the hypothalamic–pituitary–adrenal (HPA) axis, and the levels of TNF-α, IL-6, and IL-1β can effectively reflect the degree of inflammatory stress response after sleep deprivation. IL-6 and IL-1β have strong pro-inflammatory activity [40] and can induce a variety of pro-inflammatory mediators, such as cytokines and chemokines. Similar to IL-1β, TNF-α is a pleiotropic pro-inflammatory cytokine that plays different roles in regulating multiple developmental and immune processes, including inflammation, differentiation, lipid metabolism, and apoptosis [41]. The dysregulation of TNF-α is associated with a variety of pathological conditions, such as infection, autoimmune diseases, cancer, atherosclerosis, Alzheimer’s disease, inflammatory bowel disease, and intervertebral disc degeneration [42,43,44]. Based on data from animal and clinical studies, studies have shown that a subpopulation of MDD patients, both in animal models and clinical trials, exhibits increased plasma TNF-α levels, and blocking TNF-α improves depressive symptoms [45]. In this study, arecoline interventions remarkably lowered the release of TNF-α, IL-6, and IL-1β in mice, which was consistent with the above study.

Studies have shown that cells will be damaged by reactive oxygen species and respond to the damage of harmful substances through a variety of mechanisms, and the most important defense mechanism of oxidative stress is regulated through the Nrf2-Keap1 signaling pathway [46]. SD induces microglia activation and decreases the levels of Keap1 and Nrf2 (antioxidant and anti-inflammatory factors) in the hippocampus of animals [47]. The decrease in Keap1 and Nrf2 activities in the SD-induced model group was consistent with the results of this experiment. Keap1 is a negative regulator of Nrf2 activity, which can directly bind Nrf2 and regulate the NRF2-KEap1 signaling pathway. When cells are stressed, the p62 protein is phosphorylated at Ser403 and Ser351, and the phosphorylated p62 binds Keap1 through the KIR domain, resulting in Nrf2 dissociation from Keap1 and activation [48]. In addition, Keap1 restricts the actin cytoskeleton and Nrf2 to the cytoplasm by binding to them, respectively, and also plays an active role in Nrf2 ubiquitination and proteasome degradation [49]. It is shown that p62 can depolymerize the Nrf2-Keap1 complex by binding to Keap1, which plays a positive regulatory role in the NRF2-KEap1 pathway [50]. In addition, the aggregation of p62 and Keap1 has been detected in many human tumor cells, and certain neurodegenerative diseases, such as Alzheimer’s disease (AD), Parkinson’s disease (PD), and amyotrophic lateral sclerosis (ALS). They are also characterized by a large accumulation of protein aggregates in the central nervous system [51] and are all related to p62. Nrf2 is an important transcription factor that protects cells from oxidative stress. It can regulate the transcription of target genes in the nucleus, including NAD(P)H, NQO-1, and HO-1 [52], thereby avoiding apoptosis induced by oxidative stress. The results of this experiment showed that arecoline administration could significantly upregulate the expressions of Nrf2, Keap1, NQO1, P62, and HO-1 proteins, indicating that arecoline could improve the fatigued mice by regulating the Keap1/Nrf2/HO-1 pathway.

## 5. Conclusions

In summary, the current study demonstrates that arecoline has anti-fatigue activity, and the specific mechanisms are associated with elevating the levels of glucose and lipid metabolism, relieving oxidative stress damage, inhibiting neuroinflammatory response, regulating neurotransmitter levels, and modulating the Keap1/Nrf2/HO-1 signaling pathway. The current study provides new insight into the potential of arecoline in the prevention and improvement of fatigue.

## Figures and Tables

**Figure 1 nutrients-16-02783-f001:**
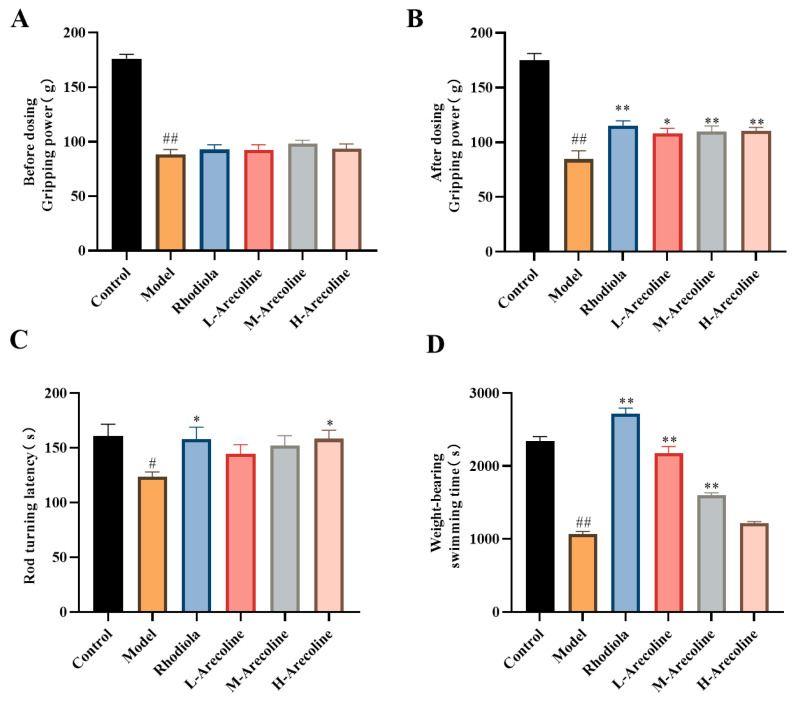
Effects of arecoline on gripping power, rod turning latency, and weight-bearing swimming time of mice. (**A**) Gripping power before dosing, (**B**) gripping power after dosing, (**C**) rod turning latency, and (**D**) weight-bearing swimming time. Data are expressed as means ± SEM (n = 12). # *p* < 0.05, ## *p* < 0.01 versus the control group. * *p* < 0.05, ** *p* < 0.01 versus the model group.

**Figure 2 nutrients-16-02783-f002:**
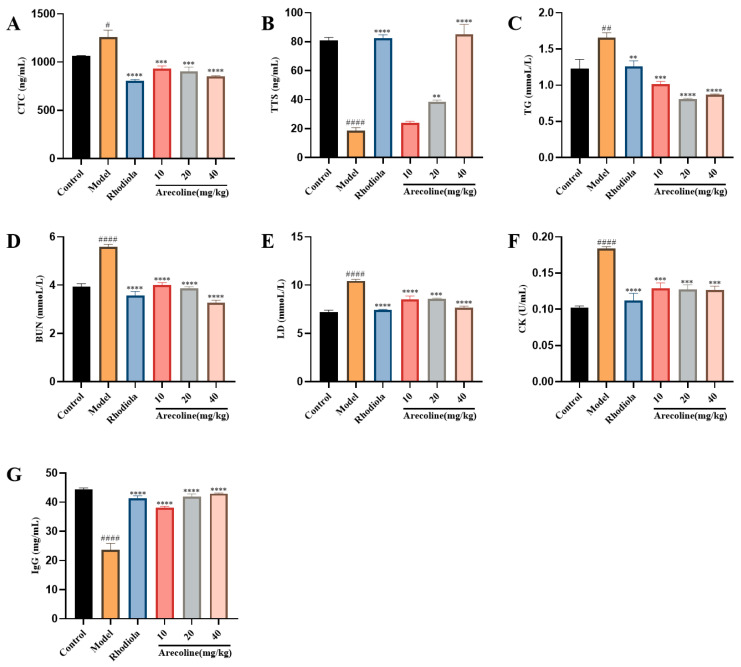
Effects of arecoline on serum biochemical indexes in mice. (**A**) CTC, (**B**) TTS, (**C**) TG, (**D**) BUN, (**E**) LD, (**F**) CK, and (**G**) IgG. Data are expressed as means ± SEM (n = 12). # *p* < 0.05, ## *p* < 0.01, #### *p* < 0.0001 versus the control group. ** *p* < 0.01, *** *p* < 0.001, **** *p* < 0.0001 versus the model group.

**Figure 3 nutrients-16-02783-f003:**
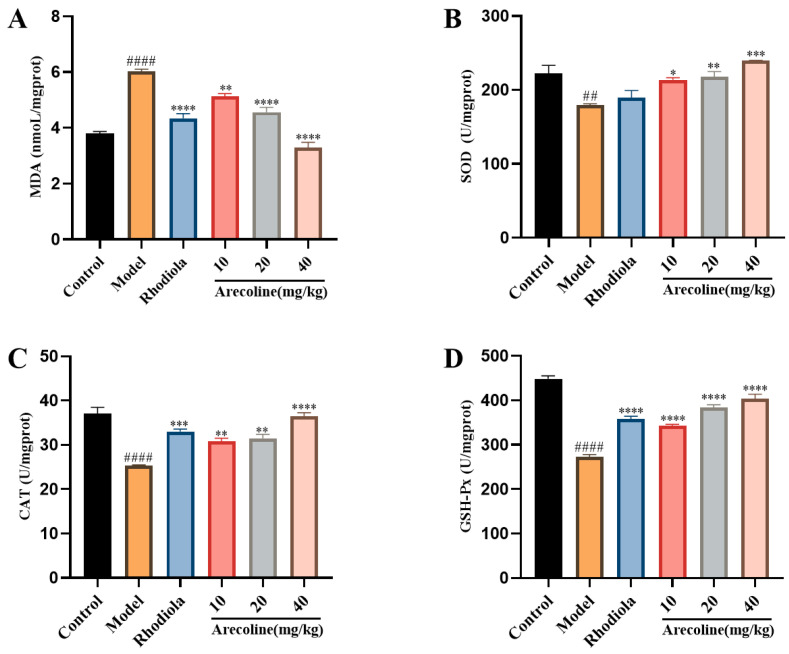
Effects of arecoline on oxidative stress indexes of the gastrocnemius muscles in mice. (**A**) MDA, (**B**) SOD, (**C**) CAT, and (**D**) GSH-Px. Data are expressed as means ± SEM (n = 12). ## *p* < 0.01, #### *p* < 0.0001 versus the control group. * *p* < 0.05, ** *p* < 0.01, *** *p* < 0.001, **** *p* < 0.0001 versus the model group.

**Figure 4 nutrients-16-02783-f004:**
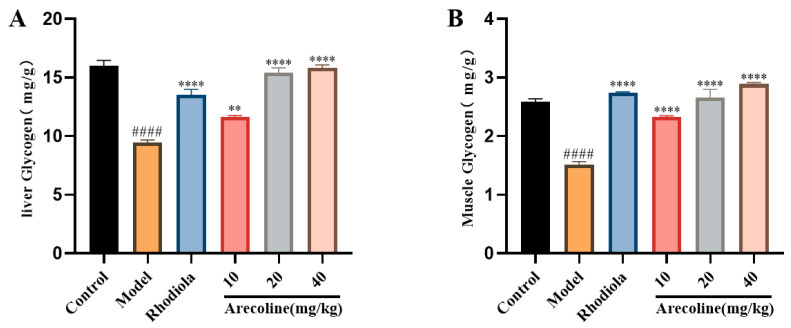
Effects of arecoline on glycolipid metabolism indexes in mice. (**A**) Liver glycogen and (**B**) muscle glycogen. Data are expressed as means ± SEM (n = 12). #### *p* < 0.0001 versus the control group. ** *p* < 0.01, **** *p* < 0.0001 versus the model group.

**Figure 5 nutrients-16-02783-f005:**
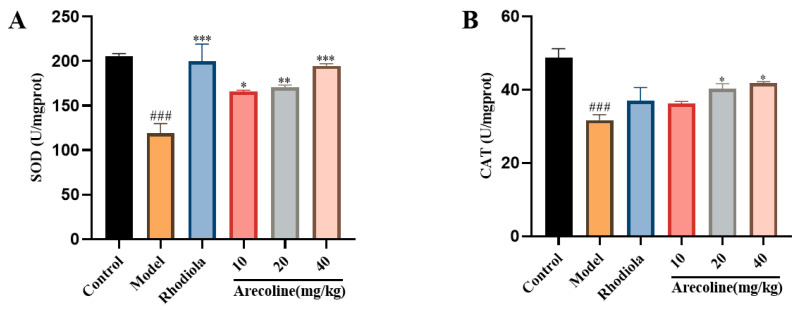
Effects of arecoline on biochemical indices of mouse hippocampus. (**A**) SOD and (**B**) CAT. Data are expressed as means ± SEM (n = 12). ### *p* < 0.001 versus the control group. * *p* < 0.05, ** *p* < 0.01, *** *p* < 0.001 versus the model group.

**Figure 6 nutrients-16-02783-f006:**
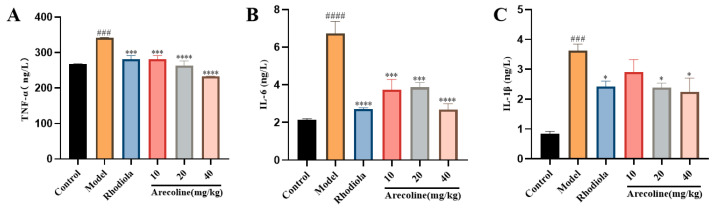
Effects of arecoline on biochemical indices of mouse hippocampus. (**A**) TNF-α, (**B**) IL-6 and (**C**) IL-1β. Data are expressed as means ± SEM (n = 12). ### *p* < 0.001, #### *p* < 0.0001 versus the control group. * *p* < 0.05, *** *p* < 0.001, **** *p* < 0.0001 versus the model group.

**Figure 7 nutrients-16-02783-f007:**
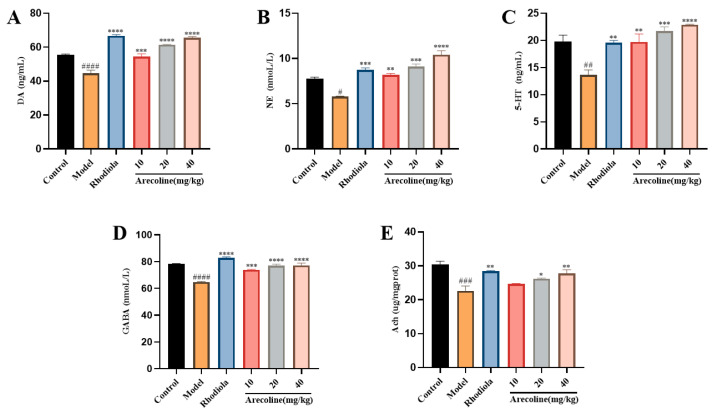
Effects of arecoline on the biochemical indices of the mouse hippocampus. (**A**) DA, (**B**) NE, (**C**) 5-HT, (**D**) GABA, and (**E**) Ach. Data are expressed as means ± SEM (n = 12). # *p* < 0.05, ## *p* < 0.01, ### *p* < 0.001, #### *p* < 0.0001 versus the Control group. * *p* < 0.05, ** *p* < 0.01, *** *p* < 0.001, **** *p* < 0.0001 versus the model group.

**Figure 8 nutrients-16-02783-f008:**
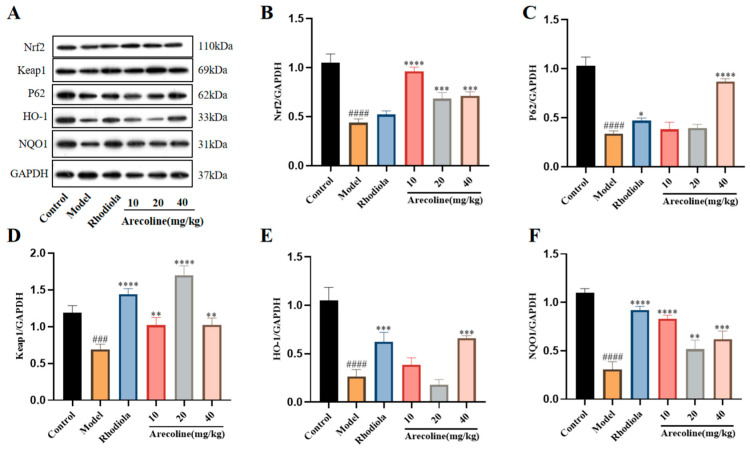
Effects of arecoline on protein expression in the gastrocnemius muscle were evaluated using western blotting. (**A**–**F**) The protein levels of Nrf2, Keap1, P62, HO-1, and NQO1 were normalized to GAPDH, and their relative band intensities were quantified. Data are expressed as means ± SD (n = 3). ### *p* < 0.001, #### *p* < 0.0001 versus the control group. * *p* < 0.05, ** *p* < 0.01, *** *p* < 0.001, **** *p* < 0.0001 versus the model group.

## Data Availability

The original contributions presented in the study are included in the article, further inquiries can be directed to the corresponding authors.

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
