# Peer review of "Research on the Anti-Fatigue Effects and Mechanisms of Arecoline in Sleep-Deprived Mice"

_nutrients, 2024, doi:10.3390/nu16162783_

Round 1

Reviewer 1 Report

Comments and Suggestions for Authors

Review for nutrients 3086012

This manuscript is poorly executed and written. It does not adhere to scientific guidelines and offers no significant contributions to the scientific field or the public. The major shortcomings are listed below:

1.         Arecoline is known for its stimulant effects and has been traditionally used in various cultures, particularly in South East Asia. However, its use as an antifatigue food supplement is highly questionable and potentially dangerous for several reasons:

A.     Addiction and Dependency.

B.     Oral Health Issues.

C.     Cancer Risk: Arecoline and areca nut chewing have been linked to an increased risk of oral and esophageal cancers. The International Agency for Research on Cancer (IARC) classifies areca nut as a Group 1 carcinogen, indicating sufficient evidence of carcinogenicity in humans. Arecoline was classified as possibly carcinogenic to humans (Group 2B) on the basis of strong mechanistic evidence.

D.     Cardiovascular Issues: Consumption of arecoline can lead to increased heart rate and blood pressure, posing risks to individuals with cardiovascular conditions.

As a result, the authors' conclusion in the abstract, stating that 'The current study provides new insight into the potential of arecoline in preventing and improving fatigue,' is misleading and could give the incorrect impression that arecoline is safe for consumption.

2.         The manuscript is poorly written, reflecting the authors' lack of attention to detail in this research. For instance, it contains different fonts and sentences with errors, such as "Error! reference source not found" in the introduction and discussion sections.

3.         The authors did not provide scientific literature on the effects of arecoline on neurotransmitters, which contribute to its psychoactive properties and potential for dependency. This omission highlights the need for a more comprehensive scientific background to fully understand the complex pharmacological actions of arecoline on neurotransmitter systems and the associated health risks.

4.         The Materials and Methods section contains numerous inaccuracies and unclear information.

A. The SD procedure is not clearly described and no reference for the positive control chosen.

B. Blood samples should be extracted from the orbital vascular plexus, not the eyeball.

C. There was no histopathological analysis reported, making it unclear why tissues were preserved in 10% formalin.

D. The description of biochemical parameter analysis in lines 190-194 is very ambiguous. The authors should specify the sample sources for each assay.

E. The Western blot analysis is misleading. It is not feasible to use 12% SDS-PAGE for target proteins ranging from 31 kDa to 110 kDa. Additionally, there is no information provided regarding the antibodies used; catalog numbers and vendors should be specified.

5.         The error bars for each assay in all figures were unusually small compared to other animal study literature, which raises concerns about the reliability of the data.

6.         The serum biochemical indexes section was poorly executed, analyzed, and lacked thorough interpretation. In line 261, the authors mentioned 'LA and BUN…' but did not clarify what 'LA' refers to. Figure 2A displayed TCT levels without explaining what 'TCT' stands for. More concerning is the LDH level: under sleep deprivation (SD) conditions, serum LDH should typically be elevated, indicating tissue damage from chronic or acute disease or injury. However, Figure 2G showed the opposite.

7.         Figure 6 shows cytokine levels in the hippocampus, but these data are not reliable as they are represented in ng/L. Tissue levels should not be presented as weight per volume.

8.         Similarly, Figure 7 biochemical indices of mouse hippocampus are not reliable with the same reason shown above.

9.         Figure 8: No labels on the lanes of Western blots. It is also unclear the blot shown was used protein samples pooled from 3 animals as n=3 or or if a single specific blot was displayed. Under physiological conditions, Keap1 protein traps Nrf2 in the cytoplasm and ubiquitinates Nrf2, thereby promoting the proteolysis of Nrf2 by proteasomes. Under stress conditions, Nrf2 dissociates from the Keap1/Nrf2 complex. Nrf2 will translocate into nucleus and activates the expression of Phase II enzymes. The authors failed to demonstrate this relationship under sleep deprivation (SD) conditions and with arecoline treatment.

Comments on the Quality of English Language

There are many ambigous descriptions.

Author Response

    Comments 1: Arecoline is known for its stimulant effects and has been traditionally used in various cultures, particularly in South East Asia. However, its use as an antifatigue food supplement is highly questionable and potentially dangerous for several reasons:

A. Addiction and Dependency.

B. Oral Health Issues 

C. Cancer Risk: Arecoline and areca nut chewing have been linked to an increased risk of oral and esophageal cancers. The International Agency for Research on Cancer (IARC) classifies areca nut as a Group 1 carcinogen, indicating sufficient evidence of carcinogenicity in humans. Arecoline was classified as possibly carcinogenic to humans (Group 2B) on the basis of strong mechanistic evidence. 

D. Cardiovascular Issues: Consumption of arecoline can lead to increased heart rate and blood pressure, posing risks to individuals with cardiovascular conditions.

As a result, the authors' conclusion in the abstract, stating that 'The current study provides new insight into the potential of arecoline in preventing and improving fatigue,' is misleading and could give the incorrect impression that arecoline is safe for consumption.

Response 1: We really appreciate your valuable advice. The incorrect description has been modified. We do not deny that arecoline has certain carcinogenic and addictive properties, which has been explained in the abstract. Our study only preliminarily confirmed that arecoline has the certain biological activity of anti-fatigue in mice, and its specific relationship in human remains to be further confirmed. Thanks again for your carefulness in reporting it.

Comments 2:.The manuscript is poorly written, reflecting the authors' lack of attention to detail in this research. For instance, it contains different fonts and sentences with errors, such as "Error! reference source not found" in the introduction and discussion sections.

Response 2: Thanks for your suggestion. We have revised and improved the related details in the text and all the references.

Comments 3: The authors did not provide scientific literature on the effects of arecoline on neurotransmitters, which contribute to its psychoactive properties and potential for dependency. This omission highlights the need for a more comprehensive scientific background to fully understand the complex pharmacological actions of arecoline on neurotransmitter systems and the associated health risks.

Response 3: Thanks very much for your warm advice. We have added the scientific literature on the effects of arecoline on neurotransmitters.(See in Page 12, Lines 408-421)

Comments 4:  The Materials and Methods section contains numerous inaccuracies and unclear information.

A. The SD procedure is not clearly described and no reference for the positive control chosen.

B. Blood samples should be extracted from the orbital vascular plexus, not the eyeball.

C. There was no histopathological analysis reported, making it unclear why tissues were preserved in 10% formalin.

D. The description of biochemical parameter analysis in lines 190-194 is very ambiguous. The authors should specify the sample sources for each assay.

E. The Western blot analysis is misleading. It is not feasible to use 12% SDS-PAGE for target proteins ranging from 31 kDa to 110 kDa. Additionally, there is no information provided regarding the antibodies used; catalog numbers and vendors should be specified.

Response 4: Thanks very much for your warm advice.

A. The SD is a model created by our laboratory, with no commonly used positive control drug reference.

B. Eyeball blood collection is an effective method for mouse blood collection in experimental animal sampling and can be used to detect relevant indicators.(Deng, Lin,et al.,2014)

C. We have removed and corrected the incorrect description.(See in Page 4, Lines 185-186)

D. We have modified the test method and labeled the samples.(See in Page 4, Lines 189-194)

E. Thank you for pointing this out. We have corrected it and provided information about the antibodies used, catalog numbers and vendors should be specified.(See in Page 3, Lines 105-106; See in Page 5, Lines 206)

Comments 5: The error bars for each assay in all figures were unusually small compared to other animal study literature, which raises concerns about the reliability of the data.

Response 5: Thanks very much for your warm advice. All our experiments used real data with good data balance, and the original data has been uploaded.

Comments 6: The serum biochemical indexes section was poorly executed, analyzed, and lacked thorough interpretation. In line 261, the authors mentioned 'LA and BUN…' but did not clarify what 'LA' refers to. Figure 2A displayed TCT levels without explaining what 'TCT' stands for. More concerning is the LDH level: under sleep deprivation (SD) conditions, serum LDH should typically be elevated, indicating tissue damage from chronic or acute disease or injury. However, Figure 2G showed the opposite.

Response 6: Thank you for pointing this out. This is our negligence and we have amended to 'LD' and 'CTC'. And we also removed incorrect descriptions and graphic information.(See in Page 6, Lines 261-270 and Page 7,Figure 2)

Comments 7: Figure 6 shows cytokine levels in the hippocampus, but these data are not reliable as they are represented in ng/L. Tissue levels should not be presented as weight per volume.

Response 7: Thanks for your suggestion. We refer to other articles knowing that weight per volume is a unit of expressed tissue level to determine the tissue level as ng/l and ng/ml.(Liang, Xin et al., 2023; Rong, Mei et al., 2024)

Comments 8: Similarly, Figure 7 biochemical indices of mouse hippocampus are not reliable with the same reason shown above.

Response 8: Thanks for your suggestion. We refer to other articles knowing that weight per volume is a unit of expressed tissue level to determine the tissue level as ng/l and ng/ml.(Liang, Xin et al., 2023; Rong, Mei et al., 2024)

Comments 9: Figure 8: No labels on the lanes of Western blots. It is also unclear the blot shown was used protein samples pooled from 3 animals as n=3 or or if a single specific blot was displayed. Under physiological conditions, Keap1 protein traps Nrf2 in the cytoplasm and ubiquitinates Nrf2, thereby promoting the proteolysis of Nrf2 by proteasomes. Under stress conditions, Nrf2 dissociates from the Keap1/Nrf2 complex. Nrf2 will translocate into nucleus and activates the expression of Phase II enzymes. The authors failed to demonstrate this relationship under sleep deprivation (SD) conditions and with arecoline treatment.

Response 9: Thanks for your suggestion. The Blot shows a single specific blot and we have added the labels on the lanes of Western blots.(See in Page 11, Figure 8).We have added relevant information, in the discussion section.(See in Page 13, Lines 444-447)

Reviewer 2 Report

Comments and Suggestions for Authors

The expression "could" (3.2, 3.3, 3.4, 3.5.2, 3.5.3) indicates a certain amount of mistrust of the experimental results. What is the cause? I recommend the authors to explain at the end of each paragraph

The conclusions are poorly presented, I suggest they be more consistent and reflect the essence of all the problems addressed.

Can it be considered that these effects will be beneficial and will have a counterpart in the case of humans?

Author Response

Comments 1: The expression "could" (3.2, 3.3, 3.4, 3.5.2, 3.5.3) indicates a certain amount of mistrust of the experimental results. What is the cause? I recommend the authors to explain at the end of each paragraph.

Response 1: We appreciated your valuable suggestion and we have improved the description of the results in each paragraph.

Comments 2: The conclusions are poorly presented, I suggest they be more consistent and reflect the essence of all the problems addressed.

Response 2: Thanks for your suggestion. We have improved the parts of the conclusions and improved their quality.(See in Page 13, Lines 464-469)

Comments 3: Can it be considered that these effects will be beneficial and will have a counterpart in the case of humans?

Response 3: Thanks very much for your warm advice. We do not deny that arecoline has certain dangers, which have been explained in the abstract.Our study only preliminarily confirmed that arecoline has certain anti-fatigue biological activity in mice, and its specific relationship in humans needs to be further confirmed.Thank you again for your reply.